# Learning word-like units from joint audio-visual analysis

**David Harwath and James R. Glass**
Computer Science and Artificial Intelligence Laboratory
Massachusetts Institute of Technology
Cambridge, MA 02139, USA
`{dharwath,glass}@mit.edu`

## Abstract

Given a collection of images and spoken audio captions, we present a method for discovering word-like acoustic units in the continuous speech signal and grounding them to semantically relevant image regions. For example, our model is able to detect spoken instances of the words "lighthouse" within an utterance and associate them with image regions containing lighthouses. We do not use any form of conventional automatic speech recognition, nor do we use any text transcriptions or conventional linguistic annotations. Our model effectively implements a form of spoken language acquisition, in which the computer learns not only to recognize word categories by sound, but also to enrich the words it learns with semantics by grounding them in images.

## 1    Introduction

### 1.1    Problem Statement and Motivation

Automatically discovering words and other elements of linguistic structure from continuous speech has been a longstanding goal in computational linguists, cognitive science, and other speech processing fields. Practically all humans acquire language at a very early age, but this task has proven to be an incredibly difficult problem for computers. While conventional automatic speech recognition (ASR) systems have a long history and have recently made great strides thanks to the revival of deep neural networks (DNNs), their reliance on highly supervised training paradigms has essentially restricted their application to the major languages of the world, accounting for a small fraction of the more than 7,000 human languages spoken worldwide (Lewis et al., 2016). The main reason for this limitation is the fact that these supervised approaches require enormous amounts of very expensive human transcripts. Moreover, the use of the written word is a convenient but limiting convention, since there are many oral languages which do not even employ a writing system. In constrast, infants learn to communicate verbally before they are capable of reading and writing - so there is no inherent reason why spoken language systems need to be inseparably tied to text.

The key contribution of this paper has two facets. First, we introduce a methodology capable of not only discovering word-like units from continuous speech at the waveform level with no additional text transcriptions or conventional speech recognition apparatus. Instead, we jointly learn the semantics of those units via visual associations. Although we evaluate our algorithm on an English corpus, it could conceivably run on any language without requiring any text or associated ASR capability. Second, from a computational perspective, our method of speech pattern discovery runs in linear time. Previous work has presented algorithms for performing acoustic pattern discovery in continuous speech (Park & Glass, 2008; Jansen et al., 2010; Jansen & Van Durme, 2011) without the use of transcriptions or another modality, but those algorithms are limited in their ability to scale by their inherent $O(n^2)$ complexity, since they do an exhaustive comparison of the data against itself. Our method leverages correlated information from a second modality - the visual domain - to guide the discovery of words and phrases. This enables our method to run in $O(n)$ time, and we demonstrate it scalability by discovering acoustic patterns in over 522 hours of audio data.

## 1.2 PREVIOUS WORK

A sub-field within speech processing that has garnered much attention recently is unsupervised speech pattern discovery. Segmental Dynamic Time Warping (S-DTW) was introduced by Park & Glass (2008), which discovers repetitions of the same words and phrases in a collection of untranscribed acoustic data. Many subsequent efforts extended these ideas(Jansen et al., 2010; Jansen & Van Durme, 2011; Dredze et al., 2010; Harwath et al., 2012; Zhang & Glass, 2009). Alternative approaches based on Bayesian nonparametric modeling (Lee & Glass, 2012; Ondel et al., 2016) employed a generative model to cluster acoustic segments into phoneme-like categories, and related works aimed to segment and cluster either reference or learned phoneme-like tokens into word-like and higher-level units (Johnson, 2008; Goldwater et al., 2009; Lee et al., 2015).

In parallel, the computer vision and NLP communities have begun to leverage deep learning to create multimodal models of images and text. Many works have focused on generating annotations or text captions for images (Socher & Li, 2010; Frome et al., 2013; Socher et al., 2014; Karpathy et al., 2014; Karpathy & Li, 2015; Vinyals et al., 2015; Fang et al., 2015; Johnson et al., 2016). One interesting intersection between word induction from phoneme strings and multimodal modeling of images and text is that of Gelderloos & Chrupaa (2016), who uses images to segment words within captions at the phoneme string level. Several recent papers have taken these ideas beyond text, and attempted to relate images to spoken audio captions directly at the waveform level (Harwath & Glass, 2015; Harwath et al., 2016).

While supervised object detection is a standard problem in the vision community, several recent works have tackled the problem of weakly-supervised or unsupervised object localization (Bergamo et al., 2014; Cho et al., 2015; Zhou et al., 2015; Cinbis et al., 2016). Although the focus of this work is discovering acoustic patterns, in the process we jointly associate the acoustic patterns with clusters of image crops, which we demonstrate capture visual patterns as well.

## 2 EXPERIMENTAL DATA

We employ a corpus of over 200,000 spoken captions for images taken from the Places205 dataset (Zhou et al., 2014), corresponding to over 522 hours of speech data. The captions were collected using Amazon's Mechanical Turk service, in which workers were shown images and asked to describe them verbally in a free-form manner. Our data collection scheme is described in detail in Harwath et al. (2016), but the experiments in this paper leverage nearly twice the amount of data. For training our multimodal neural network as well as the pattern discovery experiments, we use a subset of 214,585 image/caption pairs, and we hold out a set of 1,000 pairs for evaluating the performance of the multimodal network's retrieval ability. Because we lack ground truth text transcripts for the data, we used Google's Speech Recognition public API to generate proxy transcripts which we use when analyzing our system. Note that the ASR was only used for analysis of the results, and was not involved in any of the learning.

## 3 AUDIO-VISUAL EMBEDDING NEURAL NETWORKS

We first train a deep multimodal embedding network similar in spirit to the one described in Harwath et al. (2016), but with a more sophisticated architecture. The model is trained to map entire image frames and entire spoken captions into a shared embedding space; however, as we will show, the trained network can then be used to localize patterns corresponding to words and phrases within the spectrogram, as well as visual objects within the image by applying it to small sub-regions of the image and spectrogram. The model is comprised of two branches, one which takes as input images, and the other which takes as input spectrograms. The image network is formed by taking the off-the-shelf VGG 16 layer network (Simonyan & Zisserman, 2014) and replacing the softmax classification layer with a linear transform which maps the 4096-dimensional activations of the second fully connected layer into our 1024-dimensional multimodal embedding space. In our experiments, the weights of this projection layer are trained, but the layers taken from the VGG network below it are kept fixed. The second branch of our network analyzes speech spectrograms as if they were black and white images. Our spectrograms are computed using 40 log Mel filterbanks with a 25ms Hamming window and a 10ms shift. Therefore, the input to this branch always has 1 color channel

and is always 40 pixels high (corresponding to the 40 Mel filterbanks), but the width of the spectrogram varies depending upon the duration of the spoken caption, with each pixel corresponding to approximately 10 milliseconds worth of audio. The specific network architecture we use is shown below, where C denotes the number of convolutional channels, W is filter width, H is filter height, and S is pooling stride.

1. Convolution with C=128, W=1, H=40, ReLU
2. Convolution with C=256, W=11, H=1, ReLU, maxpool with W=3, H=1, S=2
3. Convolution with C=512, W=17, H=1, ReLU, maxpool with W=3, H=1, S=2
4. Convolution with C=512, W=17, H=1, ReLU, maxpool with W=3, H=1, S=2
5. Convolution with C=1024, W=17, H=1, ReLU
6. Meanpool over entire caption width followed by L2 normalization

In practice during training, we restrict the caption spectrograms to all be 1024 frames wide (i.e., 10sec of speech) by applying truncation or zero padding; this introduces computational savings and was shown in Harwath et al. (2016) to only slightly degrade the performance. Additionally, both the images and spectrograms are mean normalized before training. The overall multimodal network is formed by tying together the image and audio branches with a layer which takes both of their output vectors and computes an inner product between them, representing the similarity score between a given image/caption pair. We train the network to assign high scores to matching image/caption pairs, and lower scores to mismatched pairs. The objective function and training procedure we use is identical to that described in Harwath et al. (2016), but we briefly describe it here.

Within a minibatch of $B$ image/caption pairs, let $S_j^p$, $j = 1, \ldots, B$ denote the similarity score of the $j^{th}$ image/caption pair as output by the neural network. Next, for each pair we randomly sample one impostor caption and one impostor image from the same minibatch. Let $S_j^i$ denote the similarity score between the $j^{th}$ caption and its impostor image, and $S_j^c$ be the similarity score between the $j^{th}$ image and its impostor caption. The total loss for the entire minibatch is then computed as

$$\mathcal{L}(\theta) = \sum_{j=1}^{B} \max(0, S_j^c - S_j^p + 1) + \max(0, S_j^i - S_j^p + 1). \tag{1}$$

We train the neural network with 50 epochs of stochastic gradient descent using a batch size $B = 128$, a momentum of 0.9, and a learning rate of 1e-5 which is set to geometrically decay by a factor between 2 and 5 every 5 to 10 epochs.

## 4 FINDING AND CLUSTERING AUDIO-VISUAL CAPTION GROUNDINGS

Although we have trained our multimodal network to compute embeddings at the granularity of entire images and entire caption spectrograms, we can easily apply it in a more localized fashion. In the case of images, we can simply take any arbitrary crop of an original image and resize it to 224x224 pixels. The audio network is even more trivial to apply locally, because it is entirely convolutional and the final mean pooling layer ensures that the output will be a 1024-dim vector no matter the extent of the input. The bigger question is *where* to locally apply the networks in order to discover meaningful acoustic and visual patterns.

Given an image and its corresponding spoken audio caption, we use the term grounding to refer to extracting meaningful segments from the caption and associating them with an appropriate sub-region of the image. For example, if an image depicted a person eating ice cream and its caption contained the spoken words "A person is enjoying some ice cream," an ideal set of groundings would entail the acoustic segment containing the word "person" linked to a bounding box around the person, and the segment containing the word "ice cream" linked to a box around the ice cream. We use a constrained brute force ranking scheme to evaluate all possible groundings (with a restricted granularity) between an image and its caption. Specifically, we divide the image into a grid, and extract all of the image crops whose boundaries sit on the grid lines. Because we are mainly interested in extracting regions of interest and not high precision object detection boxes, to keep the number of proposal regions under control we impose several restrictions. First, we use a 10x10 grid on each image regardless of its original size. Second, we define minimum and maximum aspect ratios as 2:3

and 3:2 so as not to introduce too much distortion and also to reduce the number of proposal boxes. Third, we define a minimum bounding width as 30% of the original image width, and similarly a minimum height as 30% of the original image height. In practice, this results in a few thousand proposal regions per image.

To extract proposal segments from the audio caption spectrogram, we similarly define a 1-dim grid along the time axis, and consider all possible start/end points at 10 frame (pixel) intervals. We impose minimum and maximum segment length constraints at 50 and 100 frames (pixels), implying that our discovered acoustic patterns are restricted to fall between 0.5 and 1 second in duration. The number of proposal segments will vary depending on the caption length, and typically number in the several thousands. Note that when learning groundings we consider the entire audio sequence, and do not incorporate the 10sec duration constraint imposed during the first stage of learning.

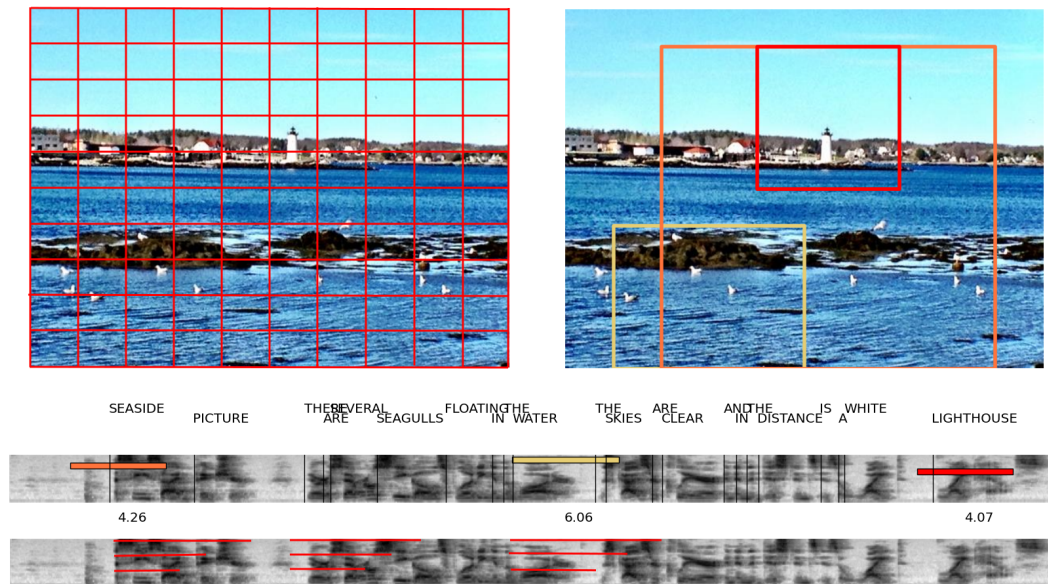

Figure 1: An example of our grounding method. The left image displays a grid defining the allowed start and end coordinates for the bounding box proposals. The bottom spectrogram displays several audio region proposals drawn as the families of stacked red line segments. The image on the right and spectrogram on the top display the final output of the grounding algorithm. The top spectrogram also displays the time-aligned text transcript of the caption, so as to demonstrate which words were captured by the groundings. In this example, the top 3 groundings have been kept, with the colors indicating the audio segment which is grounded to each bounding box.

Once we have extracted a set of proposed visual bounding boxes and acoustic segments for a given image/caption pair, we use our multimodal network to compute a similarity score between each unique image crop/acoustic segment pair. Each triplet of an image crop, acoustic segment, and similarity score constitutes a proposed grounding. A naive approach would be to simply keep the top $N$ groundings from this list, but in practice we ran into two problems with this strategy. First, many proposed acoustic segments capture mostly silence due to pauses present in natural speech. We solve this issue by using a simple voice activity detector (VAD) which was trained on the TIMIT corpus(Garofolo et al., 1993). If the VAD estimates that 40% or more of any proposed acoustic segment is silence, we discard that entire grounding. The second problem we ran into is the fact that the top of the sorted grounding list is dominated by highly overlapping acoustic segments. This makes sense, because highly informative content words will show up in many different groundings with slightly perturbed start or end times. To alleviate this issue, when evaluating a grounding from the top of the proposal list we compare the interval intersection over union (IOU) of its acoustic segment against all acoustic segments already accepted for further consideration. If the IOU exceeds a threshold of 0.1, we discard the new grounding and continue moving down the list. We stop accumulating groundings once the scores fall to below 50% of the top score in the "keep" list, or when 10 groundings have been added to the "keep" list, whichever comes first. Figure 1 displays a pictorial example of our grounding procedure.

Once we have completed the grounding procedure, we are left with a small set of regions of interest in each image and caption spectrogram. We use the respective branches of our multimodal network to compute embedding vectors for each grounding's image crop and acoustic segment. We then employ $k$-means clustering separately on the collection of image embedding vectors as well as the collection of acoustic embedding vectors. The last step is to establish an affinity score between each image cluster $\mathcal{I}$ and each acoustic cluster $\mathcal{A}$; we do so using the equation

$$\text{Affinity}(\mathcal{I}, \mathcal{A}) = \sum_{\mathbf{i} \in \mathcal{I}} \sum_{\mathbf{a} \in \mathcal{A}} \mathbf{i}^\top \mathbf{a} \cdot \text{Pair}(\mathbf{i}, \mathbf{a}) \tag{2}$$

where $\mathbf{i}$ is an image crop embedding vector, $\mathbf{a}$ is an acoustic segment embedding vector, and $\text{Pair}(\mathbf{i}, \mathbf{a})$ is equal to 1 when $\mathbf{i}$ and $\mathbf{a}$ belong to the same grounding pair, and 0 otherwise. After clustering, we are left with a set of acoustic pattern clusters, a set of visual pattern clusters, and a set of linkages describing which acoustic clusters are associated with which image clusters. In the next section, we investigate the properties of these clusters in more detail.

## 5    EXPERIMENTS AND ANALYSIS

We trained our multimodal network on a set of 214,585 image/caption pairs, and vetted it with an image search (given caption, find image) and annotation (given image, find caption) task similar to the one used in Harwath et al. (2016); Karpathy et al. (2014); Karpathy & Li (2015). The image annotation and search recall scores on a 1,000 image/caption pair held-out test set are shown in Table 1, and are compared against the model architecture used in Harwath et al. (2016). We then performed the grounding and pattern clustering steps on the entire training dataset. This resulted in a total of 1,161,305 unique grounding pairs.

In order to evaluate the acoustic pattern discovery and clustering, we wish to assign a label to each cluster and cluster member, but this is not completely straightforward since each acoustic segment may capture part of a word, a whole word, multiple words, etc. Our strategy is to force-align the Google recognition hypothesis text to the audio, and then assign a label string to each acoustic segment based upon which words it overlaps in time. The alignments are created with the help of a Kaldi (Povey et al., 2011) speech recognizer based on the standard WSJ recipe and trained using the Google ASR hypothesis as a proxy for the transcriptions. Any word whose duration is overlapped 30% or more by the acoustic segment is included in the label string for the segment. We then employ a majority vote scheme to derive the overall cluster labels. When computing the purity of a cluster, we count a cluster member as matching the cluster label as long as the overall cluster label appears in the member's label string. In other words, an acoustic segment overlapping the words "the lighthouse" would receive credit for matching the overall cluster label "lighthouse". Several example clusters and a breakdown of the labels of their members are shown in Table 2. We investigated some simple schemes for predicting highly pure clusters, and found that the empirical variance of the cluster members (average squared distance to the cluster centroid) was a good indicator. Figure 2 displays a scatter plot of cluster purity weighted by the natural log of the cluster size against the empirical variance. Large, pure clusters are easily predicted by their low empirical variance, while a high empirical variance is indicative of a garbage cluster.

Ranking a set of $k = 500$ acoustic clusters by their variance, Table 3 displays some statistics for the 50 lowest-variance clusters. We see that most of the clusters are very large and highly pure, and their labels reflect interesting object categories being identified by the neural network. We additionally compute the coverage of each cluster by counting the total number of instances of the cluster label anywhere in the training data, and then compute what fraction of those instances were captured by the cluster. We notice many examples of high coverage clusters, e.g. the "skyscraper" cluster captures 84% of all occurrences of the word "skyscraper" anywhere in the training data, while the "baseball" cluster captures 86% of all occurrences of the word "baseball". This is quite impressive given the fact that no conventional speech recognition was employed, and neither the multimodal neural network nor the grounding algorithm had access to the text transcripts of the captions.

To get an idea of the impact of the $k$ parameter as well as a variance-based cluster pruning threshold based on Figure 2, we swept $k$ from 250 to 2000 and computed a set of statistics shown in Table 4. We compute the standard overall cluster purity evaluation metric in addition to the average coverage across clusters. The table shows the natural tradeoff between cluster purity and redundancy

(indicated by the average cluster coverage) as $k$ is increased. In all cases, the variance-based cluster pruning greatly increases both the overall purity and average cluster coverage metrics. We also notice that more unique cluster labels are discovered with a larger $k$.

Next, we examine the image clusters. Figure 3 displays the 9 most central image crops for a set of 10 different image clusters, along with the majority-vote label of each image cluster's associated audio cluster. In all cases, we see that the image crops are highly relevant to their audio cluster label. We include many more example image clusters in Appendix A.

Finally, we wish to examine the semantic embedding space in more depth. We took the top 150 clusters from the same $k = 500$ clustering run described in Table 3 and performed t-SNE (van der Maaten & Hinton, 2008) analysis on the cluster centroid vectors. We projected each centroid down to 2 dimensions and plotted their majority-vote labels in Figure 4. Immediately we see that different clusters which capture the same label closely neighbor one another, indicating that distances in the embedding space do indeed carry information discriminative across word types (and suggesting that a more sophisticated clustering algorithm than $k$-means would perform better). More interestingly, we see that semantic information is also reflected in these distances. The cluster centroids for "lake," "river," "body," "water," "waterfall," "pond," and "pool" all form a tight meta-cluster, as do "restaurant," "store," "shop," and "shelves," as well as "children," "girl," "woman," and "man." Many other semantic meta-clusters can be seen in Figure 4, suggesting that the embedding space is capturing information that is highly discriminative both acoustically *and* semantically.

Table 1: Results for image search and annotation on the Places audio caption data (214k training pairs, 1k testing pairs). Recall is shown for the top 1, 5, and 10 hits. The model we use in this paper is compared against the meanpool variant of the model architecture presented in Harwath et al. (2016). For both training and testing, the captions were truncated/zero-padded to 10 seconds.

| Model | Search | | | Annotation | | |
|---|---|---|---|---|---|---|
| | R@1 | R@5 | R@10 | R@1 | R@5 | R@10 |
| (Harwath et al., 2016) | 0.090 | 0.261 | 0.372 | 0.098 | 0.266 | 0.352 |
| This work | 0.112 | 0.312 | 0.431 | 0.120 | 0.307 | 0.438 |

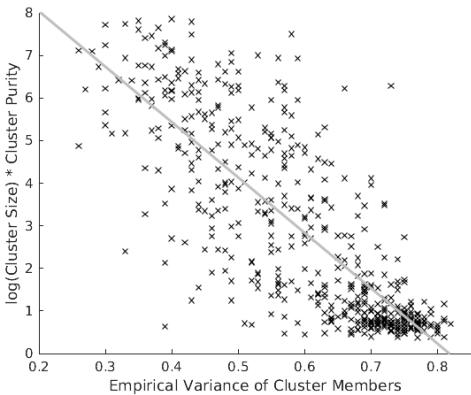

| Word | Count | Word | Count |
|---|---|---|---|
| ocean | 2150 | castle | 766 |
| (silence) | 127 | (silence) | 70 |
| the ocean | 72 | capital | 39 |
| blue ocean | 29 | large castle | 24 |
| body ocean | 22 | castles | 23 |
| oceans | 16 | (noise) | 21 |
| ocean water | 16 | council | 13 |
| (noise) | 15 | stone castle | 12 |
| of ocean | 14 | capitol | 10 |
| oceanside | 14 | old castle | 10 |

Figure 2: Scatter plot of audio cluster purity weighted by log cluster size against cluster variance for $k = 500$ (least-squares line superimposed).

Table 2: Examples of the breakdown of word/phrase identities of several acoustic clusters

## 6 CONCLUSIONS AND FUTURE WORK

In this paper, we have demonstrated that a neural network trained to associate images with the waveforms representing their spoken audio captions can successfully be applied to discover and cluster acoustic patterns representing words or short phrases in untranscribed audio data. An analogous procedure can be applied to visual images to discover visual patterns, and then the two modali-

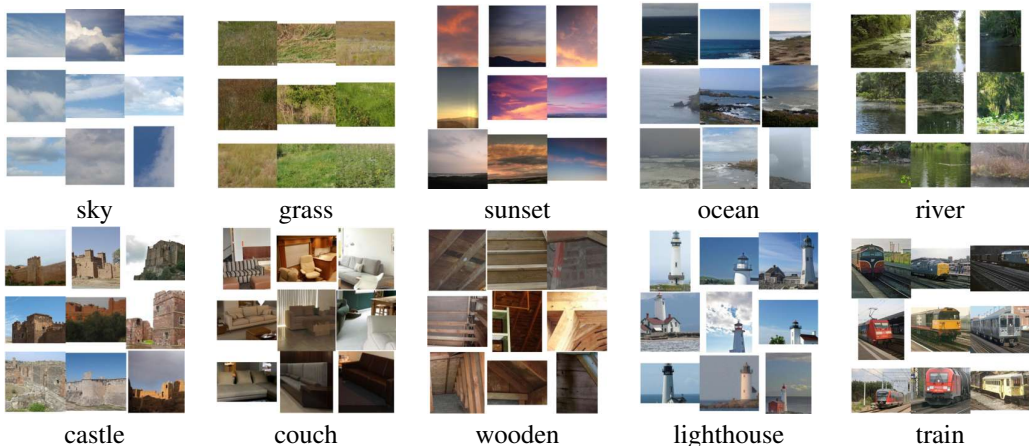

| sky | grass | sunset | ocean | river |
|---|---|---|---|---|
| castle | couch | wooden | lighthouse | train |

Figure 3: The 9 most central image crops from several image clusters, along with the majority-vote label of their most associated acoustic pattern cluster

Table 3: Top 50 clusters with $k = 500$ sorted by increasing variance. Legend: $|C_c|$ is acoustic cluster size, $|C_i|$ is associated image cluster size, Pur. is acoustic cluster purity, $\sigma^2$ is acoustic cluster variance, and Cov. is acoustic cluster coverage. A dash (-) indicates a cluster whose majority label is silence.

| Trans | $|C_c|$ | $|C_i|$ | Pur. | $\sigma^2$ | Cov. | Trans | $|C_c|$ | $|C_i|$ | Pur. | $\sigma^2$ | Cov. |
|---|---|---|---|---|---|---|---|---|---|---|---|
| - | 1059 | 3480 | 0.70 | 0.26 | - | snow | 4331 | 3480 | 0.85 | 0.26 | 0.45 |
| desert | 1936 | 2896 | 0.82 | 0.27 | 0.67 | kitchen | 3200 | 2990 | 0.88 | 0.28 | 0.76 |
| restaurant | 1921 | 2536 | 0.89 | 0.29 | 0.71 | mountain | 4571 | 2768 | 0.86 | 0.30 | 0.38 |
| black | 4369 | 2387 | 0.64 | 0.30 | 0.17 | skyscraper | 843 | 3205 | 0.84 | 0.30 | 0.84 |
| bridge | 1654 | 2025 | 0.84 | 0.30 | 0.25 | tree | 5303 | 3758 | 0.90 | 0.30 | 0.16 |
| castle | 1298 | 2887 | 0.72 | 0.31 | 0.74 | bridge | 2779 | 2025 | 0.81 | 0.32 | 0.41 |
| - | 2349 | 2165 | 0.31 | 0.33 | - | ocean | 2913 | 3505 | 0.87 | 0.33 | 0.71 |
| table | 3765 | 2165 | 0.94 | 0.33 | 0.23 | windmill | 1458 | 3752 | 0.71 | 0.33 | 0.76 |
| window | 1890 | 2795 | 0.85 | 0.34 | 0.21 | river | 2643 | 3204 | 0.76 | 0.35 | 0.62 |
| water | 5868 | 3204 | 0.90 | 0.35 | 0.27 | beach | 1897 | 2964 | 0.79 | 0.35 | 0.64 |
| flower | 3906 | 2587 | 0.92 | 0.35 | 0.67 | wall | 3158 | 3636 | 0.84 | 0.35 | 0.23 |
| sky | 4306 | 6055 | 0.76 | 0.36 | 0.34 | street | 2602 | 2385 | 0.86 | 0.36 | 0.49 |
| golf course | 1678 | 3864 | 0.44 | 0.36 | 0.63 | field | 3896 | 3261 | 0.74 | 0.36 | 0.37 |
| tree | 4098 | 3758 | 0.89 | 0.36 | 0.13 | lighthouse | 1254 | 1518 | 0.61 | 0.36 | 0.83 |
| forest | 1752 | 3431 | 0.80 | 0.37 | 0.56 | church | 2503 | 3140 | 0.86 | 0.37 | 0.72 |
| people | 3624 | 2275 | 0.91 | 0.37 | 0.14 | baseball | 2777 | 1929 | 0.66 | 0.37 | 0.86 |
| field | 2603 | 3922 | 0.74 | 0.37 | 0.25 | car | 3442 | 2118 | 0.79 | 0.38 | 0.27 |
| people | 4074 | 2286 | 0.92 | 0.38 | 0.17 | shower | 1271 | 2206 | 0.74 | 0.38 | 0.82 |
| people walking | 918 | 2224 | 0.63 | 0.38 | 0.25 | wooden | 3095 | 2723 | 0.63 | 0.38 | 0.28 |
| mountain | 3464 | 3239 | 0.88 | 0.38 | 0.29 | tree | 3676 | 2393 | 0.89 | 0.39 | 0.11 |
| - | 1976 | 3158 | 0.28 | 0.39 | - | snow | 2521 | 3480 | 0.79 | 0.39 | 0.24 |
| water | 3102 | 2948 | 0.90 | 0.39 | 0.14 | rock | 2897 | 2967 | 0.76 | 0.39 | 0.26 |
| - | 2918 | 3459 | 0.08 | 0.39 | - | night | 3027 | 3185 | 0.44 | 0.39 | 0.59 |
| station | 2063 | 2083 | 0.85 | 0.39 | 0.62 | chair | 2589 | 2288 | 0.89 | 0.39 | 0.22 |
| building | 6791 | 3450 | 0.89 | 0.40 | 0.21 | city | 2951 | 3190 | 0.67 | 0.40 | 0.50 |

ties can be linked, allowing the network to learn e.g. that spoken instances of the word "train" are associated with image regions containing trains. This is done without the use of a conventional automatic speech recognition system and zero text transcriptions, and therefore is completely agnostic to the language in which the captions are spoken. Further, this is done in $O(n)$ time with respect to the number of image/caption pairs, whereas previous state-of-the-art acoustic pattern discovery algorithms which leveraged acoustic data alone run in $O(n^2)$ time. We demonstrate the success of our methodology on a large-scale dataset of over 214,000 image/caption pairs, comprising over 522 hours of spoken audio data. We have shown that the shared multimodal embedding space learned by our model is discriminative not only across visual object categories, but also acoustically *and* semantically across spoken words. To the best of our knowledge, this paper contains by far the largest scale speech pattern discovery experiment ever performed, as well as the first ever successful effort

Table 4: Clustering statistics of the acoustic clusters for various values of $k$ and different settings of the variance-based cluster pruning threshold. Legend: $|\mathcal{C}|$ = number of clusters remaining after pruning, $|\mathcal{X}|$ = number of datapoints after pruning, Pur = purity, $|\mathcal{L}|$ = number of unique cluster labels, AC = average cluster coverage

| $k$ | $\sigma^2 < 0.9$ | | | | | $\sigma^2 < 0.65$ | | | | |
|---|---|---|---|---|---|---|---|---|---|---|
| | $|\mathcal{C}|$ | $|\mathcal{X}|$ | Pur | $|\mathcal{L}|$ | AC | $|\mathcal{C}|$ | $|\mathcal{X}|$ | Pur | $|\mathcal{L}|$ | AC |
| 250 | 249 | 1081514 | .364 | 149 | .423 | 128 | 548866 | .575 | 108 | .463 |
| 500 | 499 | 1097225 | .396 | 242 | .332 | 278 | 623159 | .591 | 196 | .375 |
| 750 | 749 | 1101151 | .409 | 308 | .406 | 434 | 668771 | .585 | 255 | .450 |
| 1000 | 999 | 1103391 | .411 | 373 | .336 | 622 | 710081 | .568 | 318 | .382 |
| 1500 | 1496 | 1104631 | .429 | 464 | .316 | 971 | 750162 | .566 | 413 | .366 |
| 2000 | 1992 | 1106418 | .431 | 540 | .237 | 1354 | 790492 | .546 | 484 | .271 |

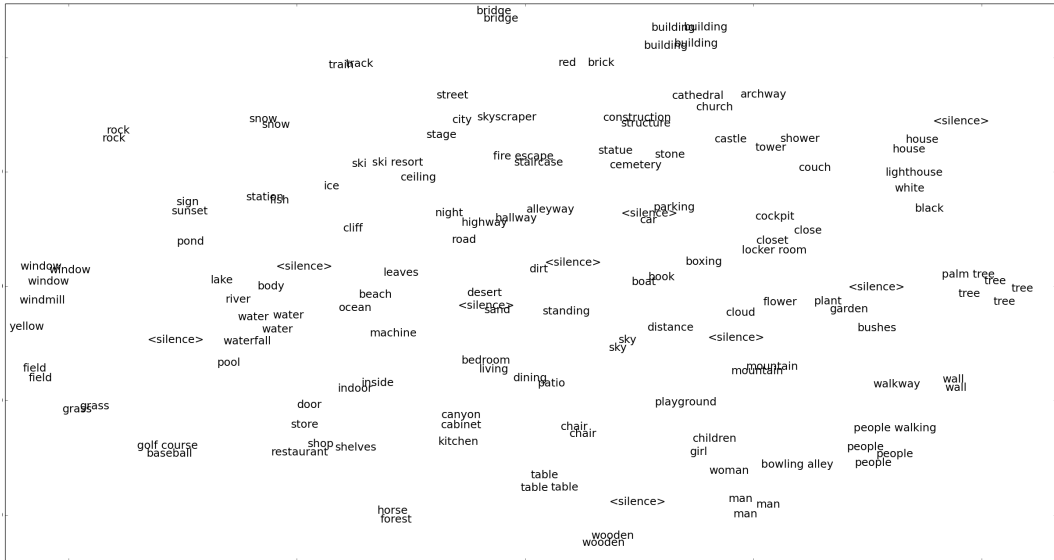

Figure 4: t-SNE analysis of the 150 lowest-variance audio pattern cluster centroids for $k = 500$. Displayed is the majority-vote transcription of the each audio cluster. All clusters shown contained a minimum of 583 members and an average of 2482, with an average purity of .668.

to learn the semantics of the discovered acoustic patterns by grounding them to patterns which are jointly discovered in another modality (images).

The future directions in which this research could be taken are incredibly fertile. Because our method creates a segmentation as well as an alignment between images and their spoken captions, a generative model could be trained using these alignments. The model could provide a spoken caption for an arbitrary image, or even synthesize an image given a spoken description. Modeling improvements are also possible, aimed at the goal of incorporating both visual and acoustic localization into the neural network itself. Additionally, by collecting a second dataset of captions for our images in a different language, such as Spanish, our model could be extended to learn the acoustic correspondences for a given object category in both languages. This paves the way for creating a speech-to-speech translation model not only with absolutely zero need for any sort of text transcriptions, but also with zero need for directly parallel linguistic data or manual human translations.

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

# A    APPENDIX: ADDITIONAL VISUALIZATIONS OF IMAGE PATTERN CLUSTERS

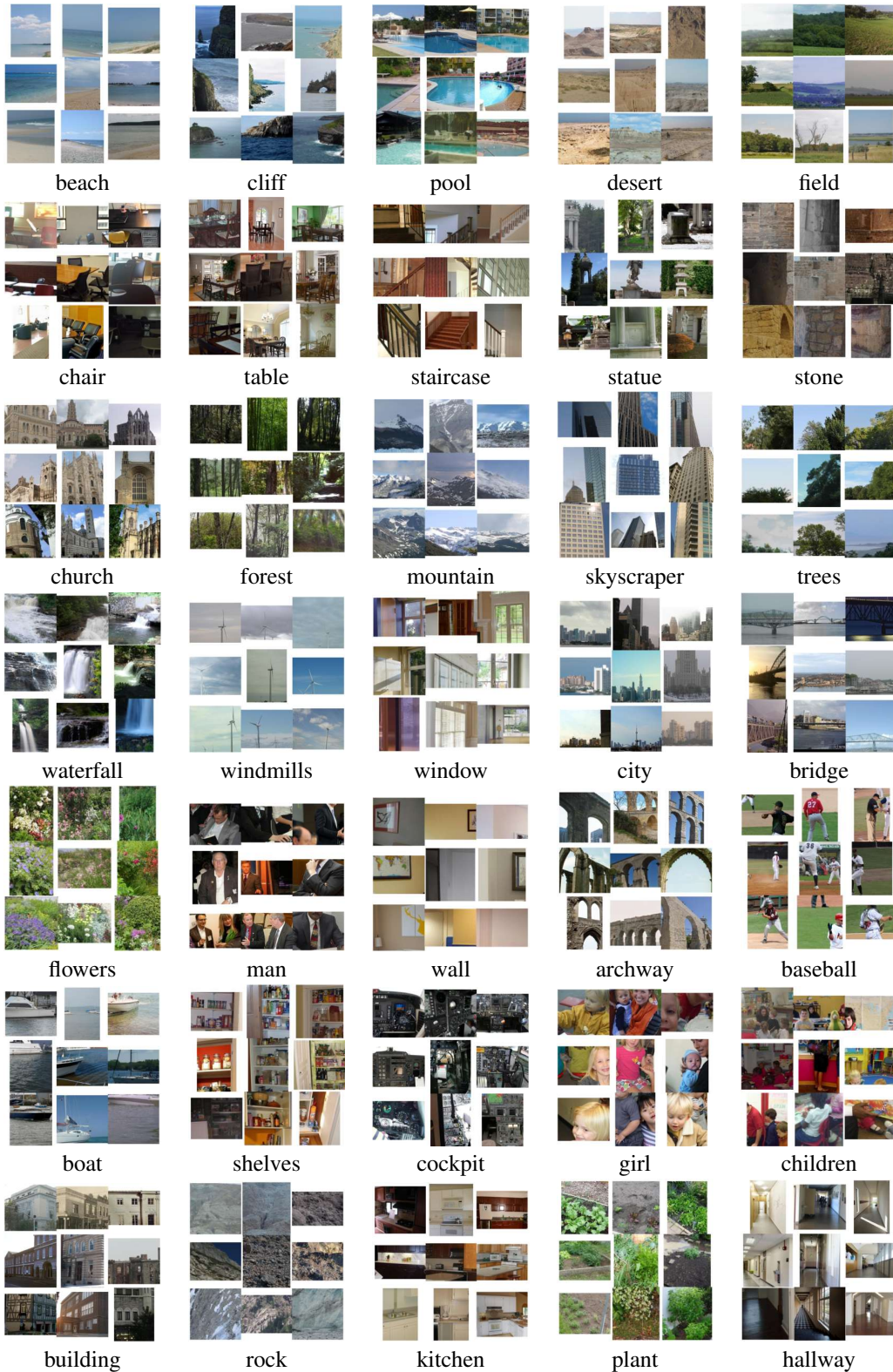

