# Peer review of "Learning Word-Like Units from Joint Audio-Visual Analylsis"

_ICLR 2017 — rejected_

[Official Review · AnonReviewer1 · rating 5 · confidence 5 · 16 Dec 2016]
**Learning word-like units from joint audio-visual analysis**

This work proposes a joint classification of images and audio captions for the task of word like discovery of acoustic units that correlate to semantically visual objects. The general this is a very interesting direction of research as it allows for a richer representation of data: regularizing visual signal with audio and visa versa. This allows for training of visual models from video, etc. 

A major concern is the amount of novelty between this work and the author's previous publication at NIPs 2016. The authors claim a more sophisticated architecture and indeed show an improvement in recall. However, the improvements are marginal, and the added complexity to the architecture is a bit ad hoc. Clustering and grouping in section 4, is hacky. Instead of gridding the image, the authors could actually use an object detector (SSD, Yolo, FasterRCNN, etc.) to estimate accurate object proposals; rather than using k-means, a spectral clustering approach would alleviate the gaussian assumption of the distributions. In assigning visual hypotheses with acoustic segments, some form of bi-partite matching should be used.

Overall, I really like this direction of research, and encourage the authors to continue developing algorithms that can train from such multimodal datasets. However, the work isn't quite novel enough from NIPs 2016.

[Official Review · AnonReviewer2 · rating 5 · confidence 4 · 16 Dec 2016]
**Review: Learning Word-Like Units from Joint Audio-Visual Analysis**

CONTRIBUTIONS 
This paper introduces a method for learning semantic "word-like" units jointly from audio and visual data. The authors use a multimodal neural network architecture which accepts both image and audio (as spectrograms) inputs. Joint training allows one to embed both image and spoken language captions into a shared representation space. Audio-visual groundings are generated by measuring affinity between image patches and audio clips. This allows the model to relate specific visual regions to specific audio segments. Experiments cover image search (audio to image) and annotation (image to audio) tasks and acoustic word discovery.


NOVELTY+SIGNIFICANCE
As correctly mentioned in Section 1.2, the computer vision and natural language communities have studied multimodal learning for use in image captioning and retrieval. With regards to multimodal learning, this paper offers incremental advancements since it primarily uses a novel combination of input modalities (audio and images).

However, bidirectional image/audio retrieval has already been explored by the authors in prior work (Harwath et al, NIPS 2016). Apart from minor differences in data and CNN architecture, the training procedure in this submission is identical to this prior work. The novelty in this submission is therefore the procedure for using the trained model for associating image regions with audio subsequences.

The methods employed for this association are relatively straightforward combination of standard techniques with limited novelty. The trained model is used to compute alignment scores between densely sampled image regions and audio subsequences; from these alignment scores a number of heuristics are applied to associate clusters of image regions with clusters of audio subsequences.


MISSING CITATION
There is a lot of work in this area spanning computer vision, natural language, and speech recognition. One key missing reference:

Ngiam, et al. "Multimodal deep learning." ICML 2011


POSITIVE POINTS
- Using more data and an improved CNN architecture, this paper improves on prior work for bidirectional image/audio retrieval
- The presented method performs efficient acoustic pattern discovery
- The audio-visual grounding combined with the image and acoustic cluster analysis is successful at discovering audio-visual cluster pairs

NEGATIVE POINTS
- Limited novelty, especially compared with Harwath et al, NIPS 2016
- Although it gives good results, the clustering method has limited novelty and feels heuristic
- The proposed method includes many hyperparameters (patch size, acoustic duration, VAD threshold, IoU threshold, number of k-means clusters, etc) and there is no discussion of how these were set or the sensitivity of the method to these choices

[Official Review · AnonReviewer3 · rating 6 · confidence 4 · 19 Dec 2016]

This paper is a follow-up on the NIPS 2016 paper "Unsupervised learning of spoken language with visual context", and does exactly what that paper proposes in its future work section: "to perform acoustic segmentation and clustering, effectively learning a lexicon of word-like units" using the embeddings that their system learns. The analysis is very interesting and I really like where the authors are going with this.

My main concern is novelty. It feels like this work is a rather trivial follow-up on an existing model, which is fine, but then the analysis should be more satisfying: currently, it feels like the authors are just illustrating some of the things that the NIPS model (with some minor improvements) learns. For a more interesting analysis, I would have liked things like a comparison of different segmentation approaches (both in audio and in images), i.e., suppose we have access to the perfect segmentation in both modalities, what happens? It would also be interesting to look at what is learned with the grounded representation, and evaluate e.g. on multi-modal semantics tasks.

Apart from that, the paper is well written and I really like this research direction. It is very important to analyze what models learn, and this is a good example of the types of questions one should ask. I am afraid, however, that the model is not novel enough, nor the questions deep enough, to make this paper better than borderline for ICLR.

[Author Response · David Harwath · 14 Jan 2017]
**Rebuttal to reviews**

We'd like to first thank the reviewers for taking the time to read our submission and offer their thoughtful opinions and encouragement. Since the three reviewers raise some similar questions and concerns, I'll try to address them in one post instead of multiple replies.

I'll first address the issue of novelty, which is the main criticism raised by all three reviewers. When thinking about research, I think it's important to consider the problem to be solved separately from the tool(s) used to solve it. While I agree that the deep neural network architecture used in this submission is not a significant departure from the one used in our NIPS 2016 paper, the way we use it is very different. I believe that the problem we address in this submission - that is, joint localization/isolation, clustering, and association of word-like speech patterns and object-like visual patterns - is significantly novel and wasn't studied in our NIPS paper. I don't believe that this submission qualifies as just an analysis paper, because it actually attempts to solve a specific and novel problem, and does so surprisingly well.

There exists an active sub-field of the speech/linguistics/cogsci communities which studies the problem of acquiring language from untranscribed speech audio alone (see relevant citations in the submission), and most of the techniques used in that problem space rely on segmentation and clustering of the speech signal into linguistically meaningful units (often called unsupervised term discovery or UTD when the desired granularity of the units is at the word or phrase level). The most widely-used and successful techniques for UTD are based on segmental dynamic time warping, which is inherently O(N^2) complexity. I believe that one of the most significant contributions of this submission is the demonstration that the addition of contextually relevant visual information is sufficient to reduce the computational complexity of UTD to O(N), allowing it to scale to much larger datasets.

I'd also like to respond to a few specific points:

From reviewer 2:
"As correctly mentioned in Section 1.2, the computer vision and natural language communities have studied multimodal learning for use in image captioning and retrieval. With regards to multimodal learning, this paper offers incremental advancements since it primarily uses a novel combination of input modalities (audio and images)."

I respectfully disagree that the move from text to speech audio constitutes an incremental step. The field of automatic speech recognition research has been grappling with the problem of mapping speech audio to symbolic strings for over 65 years, so it's not a trivial problem. By marrying language and vision at the raw signal level as we are here, we're not just creating models that learn the associations between words and images, but actually forcing the models to simultaneously learn how to perform speech recognition in their own, non-symbolic way.

From reviewer 1:
"Clustering and grouping in section 4, is hacky. Instead of gridding the image, the authors could actually use an object detector (SSD, Yolo, FasterRCNN, etc.) to estimate accurate object proposals; rather than using k-means, a spectral clustering approach would alleviate the gaussian assumption of the distributions. In assigning visual hypotheses with acoustic segments, some form of bi-partite matching should be used."

I think that these optimizations, while good suggestions if one's goal is to squeeze every last bit of performance out of the system, are not crucial to the central theme of the paper. We chose not to use an off-the-shelf object detection system for the same reason that we chose not to give the system oracle word boundaries derived from a speech recognizer. We wanted the model to learn to localize, identify, and associate spoken words and visual objects without being explicitly trained to do so. Spectral clustering is an O(N^3) complexity algorithm, and our problem deals with clustering on the order of a million points; a full similarity matrix at floating point precision would require on the order of 4 terabytes of memory, and that's just an O(N^2) subroutine in the spectral clustering algorithm. I realize that various approximate algorithms could possibly be made to work, but in our case I think that the fact that a classic algorithm like k-means works so well is actually a testament to the separability of the embeddings that are being learned by the multimodal CNN.

[Final Decision · Program Chairs · 06 Feb 2017]
**ICLR committee final decision**

This paper received borderline reviews. All reviewers as well as AC agree that the authors pursue a very interesting and less explored direction. The paper essentially addresses the problem of double grounding; visual information helping to group acoustic signal into words, and words helping to localize object-like regions in images. While somewhat hidden under the rug, this is what makes the paper different from the authors' previous work. The reviewers mentioned this to be a minor contribution. The AC agrees with the authors that this is an interesting and novel problem worth studying. However, the AC also agrees with the reviewers that this major novelty over the previous work is missing technical depth. The AC strongly encourages the authors to improve on this aspect of the paper. A simple intuition would be to look at